# STRyper: A macOS application for microsatellite genotyping and chromatogram management

**Jean Peccoud** *

Laboratoire Écologie et Biologie des Interactions, Équipe Écologie Évolution Symbiose, Université de Poitiers, UMR CNRS 7267, Poitiers, France

* jeanpeccoud@gmail.com

## Abstract

Microsatellite markers analyzed by capillary sequencing remain useful tools for rapid genotyping and low-cost studies. This contrasts with the lack of a free application to analyze chromatograms for microsatellite genotyping that is not restricted to human genotyping. To fill this gap, I have developed STRyper, a macOS application whose source code is published under the General Public License. STRyper only uses macOS libraries, making it very lightweight, responsive, and behaving like a modern application. Its three-pane window enables easy management and viewing of chromatograms imported from FSA and HID files, the creation of size standards and of microsatellite marker panels (including bins). STRyper features powerful search capabilities (with smart folders) and a modern graphical user interface allowing, among others, the manual correction of DNA ladders and of individual genotypes by drag-and-drop. It also introduces a new way to mitigate the effect of variations in electrophoretic conditions on estimated allele sizes.

## Introduction

More than three decades after their first use, microsatellites markers, also known as short tandem repeat (STR) loci, remain popular DNA markers to assess gene flow, population history, structure and membership, ancestry, or the integrity of laboratory breeding lines, among other uses [1,2]. When locus-specific variation is not the focus of a study, a limited number of microsatellite markers are sufficient to assess evolutionary processes affecting the whole genome and to genetically identify an individual [3]. This ability stems from the sheer number of alleles per marker, which often counts in the dozens, leading to a per-locus information amount that exceeds that of single-nucleotide polymorphisms (SNPs) [4].

Due to frequent indels affecting the number of microsatellite repeat motives, microsatellites alleles essentially differ in their length, which can be estimated by simple electrophoresis of amplicons. Amplicon sequencing by the Illumina technology has however emerged as relatively affordable and more reliable alternative to capillary electrophoresis [5–7]. In species for which tried and tested microsatellite multiplexes exist, microsatellite genotyping

**Data availability statement:** The application and source code are available at https://github.com/jeanlain/STRyper/.

**Funding:** Part of this work was funded by intramural funds from the CNRS and the University of Poitiers and by Agence Nationale de la Recherche grant ANR-20-CE02-0004 (SymChroSex). the funders had no role in study design, data collection and analysis, decision to publish, or preparation of the manuscript.

**Competing interests:** The author has declared that no competing interests exist.

via electrophoresis still offers a compelling money- and time-saving solution. At a few dollars per individual in terms of consumables (for a couple of multiplexes typically combining 10–20 loci) genotyping can be performed locally in one day, as it amounts to DNA extraction, PCR, amplicon dilution and placing a plate in a capillary sequencer. When a quick answer is needed or when only few individuals need analyzing, typically for simple genotype checking, this traditional technique remains the cheapest and easiest one.

However, the difficulty sharply rises when it comes to analyzing the results of capillary electrophoresis. As opposed to genotyping via NGS, which is generally done via fully- or partially automated free tools (e.g., [8,9]), traditional microsatellite genotyping requires inspecting fluorescence curves, therefore applications with a complex graphical user interface (GUI), which are rarely free. To various degrees, these applications are focused on human identification by genotyping and forensics. A such, they are packed with features and safeguards that are of little relevance to most researchers, which somewhat complicate their use, and which may come at a high price.

This is the case of GeneMapper by ThermoFisher Scientific, a commercial application running on the Windows operating system, and which remains, to my knowledge, the most widely used for microsatellite genotyping. A Google scholar search for "genemapper", excluding references, patents and review articles, and limited to 2023 and 2024, returned 2820 results as of July 20th 2024. Most results pertained to medicine and forensics or may correspond to preprints, but the first 130 results comprised 15 English-written studies on non-human species using traditional microsatellite analyzes, indicating that this technique is far from abandoned.

GeneMarker by Softgenetics is a similar commercial application. The price of a license of either software may restrict its installation to a single computer per research laboratory. A free alternative from ThermoFisher Scientific, Peak Scanner, has limited functionalities. Complementary command-line tools [10,11] provide missing features such as allele scoring via binning, but may dissuade those who seek to conduct fragment analyses from chromatogram import to the export of individual genotypes in a single user-friendly application. In that regard, Geneious Prime and its microsatellite analysis plugin may represent an interesting trade-off between price and features. The cost of a subscription to Geneious Prime may still appear excessive to users who do not need the features that this product offers for the analysis of DNA sequences.

Osiris [12,13], stands out as being a free, feature-rich and multi-platform (Windows and macOS) tool for STR analysis. Yet, this software is, as far as I know, rarely used by population geneticists, possibly because it is highly specialized for human identification.

Researchers, especially population geneticists, would therefore benefit from a free application enabling quick microsatellite genotyping and management of thousands of samples. To meet this need, I have developed STRyper, an open-source, lightweight and user-friendly application that can analyze chromatogram files for STR genotyping. STRyper is published under the GNU General Public License v. 3 and its name is a portmanteau of "STR" and "Genotyper". As described below, STRyper features a modern GUI allowing, among others, unconstrained chromatogram management via nested folders, advanced and dynamic metadata-based chromatogram search with "smart" folders, easy folder import/export, chromatogram and genotype filtering based on multiple criteria, the definition of microsatellite multiplexes and custom size standards, fast and responsive visualization of fluorescence curves with animated zooming and automatic vertical scaling, the manual correction of DNA ladders and of individual genotypes by drag-and-drop, and a new way to mitigate the effect of variations in electrophoretic conditions on estimated allele sizes. The application and its codebase are available at https://github.com/jeanlain/STRyper.

## Description of the application

### General characteristics and development

STRyper is designed to manage and display chromatograms generated by Applied Biosystems capillary sequencers. The application also allows managing microsatellite markers and size standards, which are required for fragment size estimation and genotyping. All functionalities of the application are driven by its GUI. As opposed to command line tools, GUI development relies on application programming interfaces and frameworks that depend on the target operating system and development tools. These were dictated by my use of the Mac operating system (macOS) and by the fact that developing STRyper was a hobby project of an evolutionary biologist, not the effort of a team of professional developers. Being unencumbered by cross-platform development gave me the freedom to choose the right tools to program a GUI that was intuitive, responsive and consistent with "native" macOS applications. STRyper was thus developed using Xcode and frameworks provided by Apple (https://developer.apple.com/library/archive/documentation/MacOSX/Conceptual/OSX_Technology_Overview/System-Frameworks/SystemFrameworks.html). These frameworks include "Core Data", which is used to define and manage objects representing chromatograms, microsatellite marker, bins, alleles, genotypes and size standards, and to save them in a persistent relational database (S1 Text). Internally, Core Data relies on the SQLite database engine to manage the persistent store. GUI elements (windows, views, controls and so on) are implemented using "AppKit". "Core Graphics" functions are used to draw fluorescent curves. "Core Animation" layers accelerate compositing via the graphical processing unit (GPU) and provide fluid animation of the interface. These object-oriented frameworks (except Core Graphics) required the use of the Objective-C programming language (a superset of C) when the project started. The application code was written in the latest version (2.0) of this language.

STRyper runs under macOS version 10.13 or higher. The application does not include third-party libraries and does not require special installation steps. Its bundle contains binaries compiled for the X86 and arm64 architectures and weighs less than 15 Megabytes, including the user guide.

### Overview of the interface

The application comprises a main window (Fig 1) composed of three panes, a design paradigm used by several database-management applications like email clients. The left collapsible sidebar is a hierarchical list of folders and subfolders containing samples, each representing an imported chromatogram file. Folder and samples can be organized freely by drag and drop. A middle pane shows the content of the selected folder (samples and associated genotypes) and comprises tabs to manage size standards and microsatellite markers. The right pane shows the traces (fluorescent curves) of selected samples and genotypes.

STRyper uses very few modal panels or dialogs to validate user actions and all actions that affect the database can be undone. Most can be achieved in a couple of clicks or less as they do not require opening and closing windows. Drag and drop can be used throughout: from importing samples to applying size standards, marker panels (multiplexes), and to manually attributing alleles or size molecular ladder fragments to peaks.

STRyper can import FSA files (HID file support is experimental, as the HID format specifications are not public) containing data for 4 or 5 channels (fluorescent dyes). Samples are imported into folders, and they can be moved or copied between folders at any time. A folder and all its content, including subfolders, samples, genotypes at microsatellite markers, associated marker panels (including bins) and custom size standards, can be archived and transferred between instances of the application. Upon importing an archived folder, any marker

Sample table

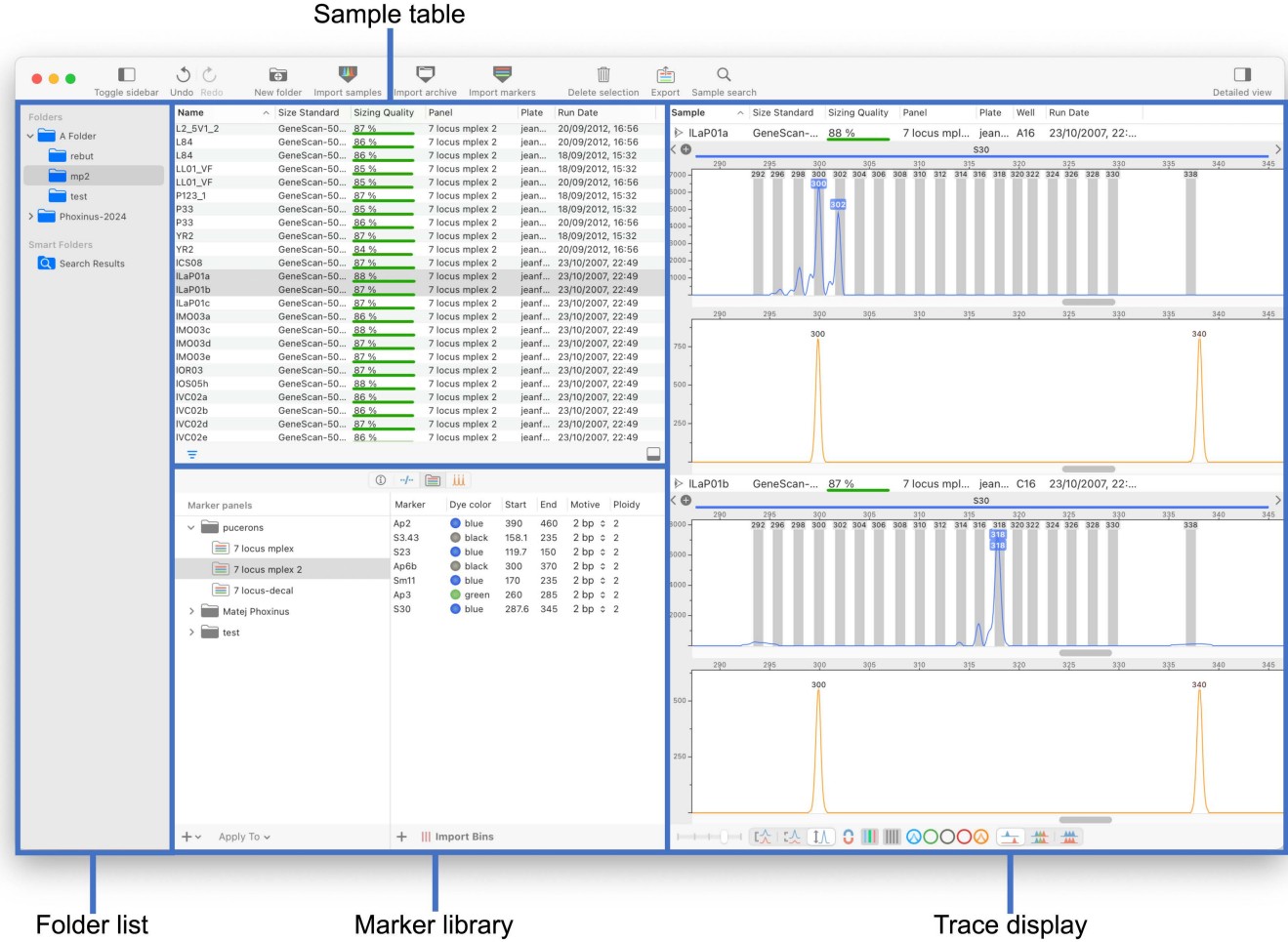

Folder list             Marker library                    Trace display

**Fig 1. The main window of STRyper.** The left pane contains the list of folders and smart folders (search results) containing samples. The middle pane is a split view comprising a top pane listing the samples of the selected folder. Its bottom pane has four tabs, which are from left to right: an inspector showing data on selected samples (Fig 2), a table of genotypes from the samples shown on the top pane, the marker library (currently shown) and the size standard library. The right pane shows the traces of selected samples in a scrollable view that can display thousands of traces.

panel and size standard encoded in the archive is imported unless is it already in the database. The imported folder therefore shows the same content as the original one.

Since samples are not constrained to compartmentalized projects, the application provides search tools to find and gather samples from the whole database. Users can define various search criteria, including sequencer run date, sizing quality, well identifier, plate name, marker panel name, etc. Search results appear in "smart folders" which dynamically update their contents as new samples meet the search criteria.

## Chromatogram display

Like all applications displaying chromatograms generated by capillary sequencers, STRyper draws plots in which the Y-axis is the fluorescence level. The X-axis represents the length of DNA fragments that produced peaks in the fluorescence. This contrasts with Osiris, in which the X-axis is the number of the fluorescence data record (the "scan" number), hence the time at which the measure was taken during the electrophoresis.

For fragment size estimates, the application must first identify fluorescence peaks, which are induced by DNA fragments. This task is performed during chromatogram import by a simple algorithm. This algorithm (detailed in the S1 Text) determines whether the fluorescence level at a given scan is elevated enough, both relative to neighbor scans and in absolute level. Peak delineation serves as a basis to subtract baseline fluorescence level, which helps peak visualization. The method developed for this task adjusts the height (fluorescence level) of a curve such that the start and end point of each peak are placed at level zero (S1 Text). Although this adjustment cannot be applied on signals that are too faint to contain meaningful peaks, it has the benefit of offering two baseline subtraction modes: one that preserves absolute peak height, and one that maintains relative peak elevation compared to the baseline (S1 Text). As this method reduces background noise, no smoothing algorithm was implemented.

Because chromatograms contain fluorescence data from several wavelengths (channels), multichannel fluorescence analysis requires determining whether a peak represents a DNA fragment or interference from another channel (i.e., "crosstalk"). The method developed for this task compares the position, shape and relative size of peaks between channels, accounting for saturation of the sequencer camera (S1 Text). To signal crosstalk to the user, the area underneath an artefactual peak is filled with the color that represents the channel that induced crosstalk (this option can be disabled). While certain applications alter fluorescence data to correct for pull-up due to crosstalk [13], flagging peaks resulting from crosstalk and leaving the source signal untouched was considered sufficient. These peaks are simply ignored in automatic detection of alleles and DNA ladder fragments (detailed below), although the user can manually assign these peaks, should they wish to.

Upon selecting samples in the table, corresponding fluorescent curves (traces) are instantaneously displayed on the right pane (Fig 1). As the application fully supports the dark theme of macOS (version 10.14 or more recent), it can display traces on a dark background to alleviate eye strain. Any region in which a peak saturated the sequencer camera is shown behind curves as a rectangle whose color reflects the channel that likely caused saturation. Traces can be scrolled and zoomed in/out horizontally via trackpad gestures such as swipe, pinch and double tap, via the scroll wheel, or by clicking/dragging the mouse over horizontal rulers. Dragging the mouse over the vertical ruler sets the fluorescence level at the top of the view, hence the vertical scale. Zooming is animated, which helps users keep track of the range (in base pairs) that is displayed.

Viewing options include automatic vertical scaling to the highest visible peaks, synchronizing the vertical scales and horizontal positions, showing/hiding region of fluorescence saturation, stacking curves from several samples or channels in the same view, and subtracting the baseline fluorescence level.

## Size standards and molecular ladders

To estimate the size of DNA fragments, a molecular ladder containing fragments of know lengths (defining a "size standard"), and tagged with a specific fluorescent dye, is added to every sample before electrophoresis. DNA ladder fragments induce peaks in the trace of the corresponding channel. To associate peak to sizes, samples must be assigned the adequate size standard. STRyper comes with several widely used size standards, namely those from the GeneScan brand. Users can easily edit these size standards within the application and make their own. They can be assigned to samples during chromatogram import (based on metadata encoded in the file) or manually. Assigning a size standard automatically triggers the detection of DNA ladder fragments in the sample.

The method used to detect DNA ladder fragments and assign them to sizes of a known size standard is based on relative peak positions and accounts for non-linear relationship between fragment size and migration speed (S1 Text). Peaks resulting from crosstalk or whose height are unusual compared to others are ignored. To account for non-linearity, a polynomial of the first, second, or third degree (depending on the user choice) is used to estimate fragment size, where the response variable is the size of a fragment specified in the size standard, and the explanatory variable is the scan number at the tip of the corresponding peak (representing migration speed). This principle is also implemented in other applications such as GeneMapper. Fitting is achieved via the Cholesky decomposition implemented in the Linear Algebra Package (https://netlib.org/lapack/). Fitting parameters are used to draw traces by computing the size in base pairs corresponding to every scan. The horizontal distance between successive scans varies unless a polynomial of the first degree (linear regression) is used for the sizing.

To evaluate the quality of the sizing, a score from 0 to 1 was developed, based on the residuals of the fitted model (differences between fragment sizes as defined in the size standard, and fragment sizes estimated by the model). This score involves computing the difference in residuals for every pair of adjacent peaks and is computed as follows. If $\Delta R$ is the difference between residuals of every pair of adjacent peaks, $\Delta S$ the difference in scan number of these peaks, $n_p$ the number of peaks and $n_s$ is number of sizes in the size standard, the quality score is:

$$1 - \max\left(\frac{\Delta R^2}{|\Delta S|}\right)\frac{10}{3} - \frac{n_s - n_p}{10}$$

Any negative score is set to zero. This formula was tuned by testing many chromatograms. The 10/3 coefficient ensures that the score is greatly reduced (often to zero) by a single assignment error, which affects $\max\left(\frac{\Delta R^2}{|\Delta S|}\right)$. A poor score compels the user to rectify the error. The score is also reduced if certain sizes of the size standard are not assigned to any peak, which increases $n_s - n_p$. A weight of 1/10 was attributed to this component, because such issue generally reflects problems during electrophoresis, which cannot be fixed in the application. Sizing quality is shown for each sample in a dedicated column displaying a gauge (Fig 1). If molecular sizing failed, sizes are not displayed on the X-axis of the chromatogram, but traces can still be viewed.

STRyper displays the trace of the molecular ladder like any other trace, letting users switch quickly between genotype and molecular ladder editing. Sizes attributed to molecular ladder fragment can be changed by dragging and dropping size labels onto peaks. Any change to the molecular ladder automatically updates the sizing of the sample without user validation. The red component of the color used for size labels is proportional to the difference between the computed size of a peak and its theoretical size, making size assignment errors easy to spot.

The application also features an inspector panel that dynamically updates to show metadata of selected samples, and most importantly, sizing information (Fig 2). This inspector can help to find sizing errors if points deviate from the curve representing the relationship between scan number and peak size.

## Microsatellite marker and bins

Genotyping requires associating chromatograms with the microsatellite markers that were amplified using fluorescent primers. Markers amplified together by multiplex PCR are

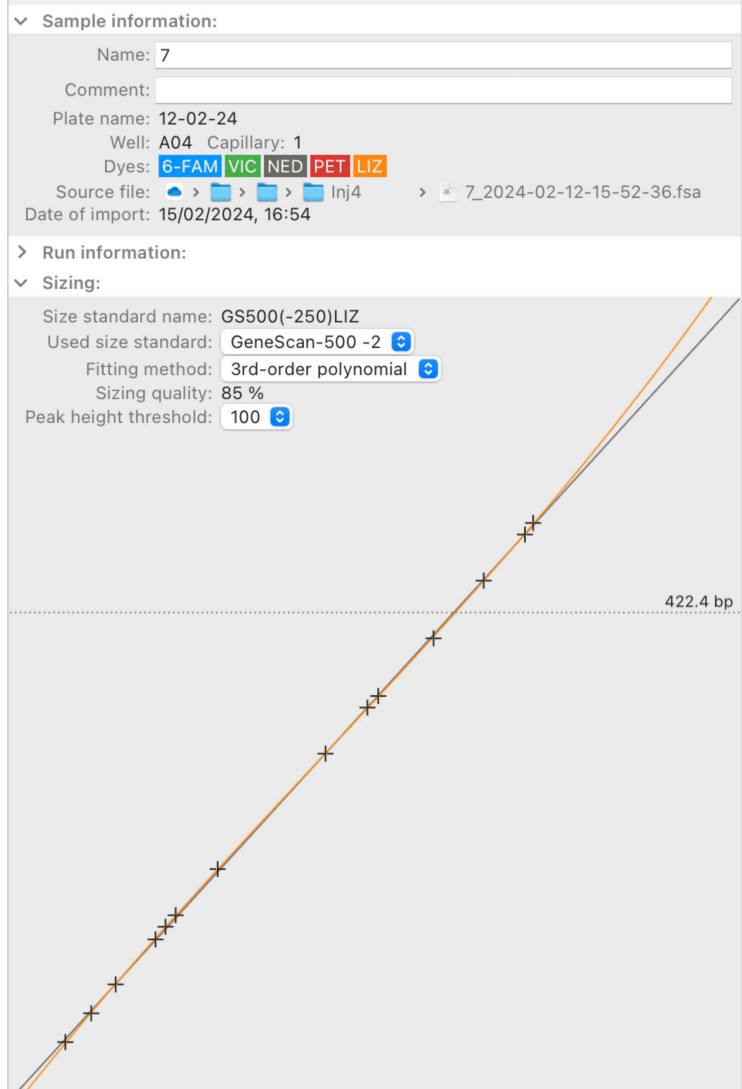

**Fig 2. The sample inspector of STRyper.** This panel with three collapsible sections dynamically updates to display information on samples that are selected in the sample table (Fig 1). The plot at the bottom shows the relationship between the time at which DNA fragments of the molecular ladder (black crosses) were detected by the sequencer camera (the X axis) and their observed sizes in base pairs (the Y axis). The relationship used to estimate fragment sizes is established by fitting a polynomial (here, of the third degree) to the points shown on the plot. This polynomial is represented by the orange curve. The horizontal dotted line indicates the estimated size at the location of the mouse cursor (cursor not shown).

regrouped into a "panel" (a term derived from GeneMapper). Users can define their own panels of haploid or diploid microsatellite markers within STRyper and organize them in folders. Markers are defined by their fluorescent dye, ploidy, length of repeat motive, name, and the size range of their alleles. These attributes can be changed after a marker is created, except for the first two. Markers can be copied and dragged between panels. Users can export marker panels to text files conforming to simple specifications described in the user guide. These text files can be imported back as marker panels. STRyper can also import panel description text files exported from GeneMapper.

A marker can comprise "bins", which are non-contiguous intervals delimiting the expected sizes of fragments corresponding to alleles [14]. Bins address the fact that estimated fragment sizes slightly vary between sequencer runs [15]. Proper bin definition must account for factors affecting amplicon mobility during electrophoresis [16], which often cause the estimated distance between consecutive microsatellite alleles (in base pairs) to slightly differ from the repeat motive length [14]. Binning can be left to specialized programs like Tandem [17], which can work on allele sizes estimated by other programs like STRyper. The management of bins within STRyper was still considered a necessity. Indeed, visualizing bins as vertical rectangles behind traces helps to characterize alleles that do not conform to the periodicity of the repeat motive, and to mitigate variations in fragment sizes between sequencer runs (further discussed below). STRyper therefore allows importing bin sets as text files (produced by GeneMapper or Tandem), but also generating and editing bin sets within the application.

In STRyper, a set of automatically named bins for a marker can be added by specifying the width and spacing of bins. To accommodate the fact that the observed distance between microsatellite alleles slightly differs from the repeat motive [14], the position and a spacing of bins can be adjusted by respectively dragging and resizing the whole bin set. Individual bins can also be added and modified via click and drag. These actions do not involve dedicated windows or panels, they can be performed at any time on the trace views where bins are displayed (Fig 1, right).

The width of a bin might not cover the full range of estimated sizes of amplicons from given allele over all electrophoretic conditions. Regularly, a peak representing an allele would fall outside the corresponding bin, although identical fragments that migrated in other sequencer runs were properly binned. To circumvent the issue, a mixture of amplicons of known sizes for each marker, known as "allelic ladder" or "inter-lane standard", can be added alongside samples for each run or sequencing plate. Allelic ladders are however only available for model species.

STRyper implements a novel approach to mitigate this issue. Rather than moving bins to match peak positions (which requires maintaining several sets of bins per marker), this approach considers that it is the estimated sizes of peaks (in base pairs), not the position of bins, which should be adjusted. The method thus correct fragment sizes using the formula $y = a + bx$, where $x$ is the size of a DNA fragment that is estimated by the DNA ladder via the fitted model mentioned earlier, $y$ is the adjusted size, and $a$ and $b$ are constants (hereafter called "offset parameters"). This approach assumes that the effect of varying electrophoresis conditions can be compensated by this linear combination. If there is no correction, $a = 0$ and $b = 1$. Good offset parameters are those that minimize the distance (in base pairs) between peaks and their corresponding bins. Because automatically determining which bins and peaks to associate might have been error-prone, a manual GUI-based method was developed. The application lets the user move and/or resize a rectangle representing the range of the bin set such that bins coincide with peaks (Fig 3). To infer offset parameters $a$ and $b$ from this operation, we let $s$ represents the start of a bin and $e$ its end, in base pairs. If $s'$ and $e'$ represent the corresponding boundaries after the user has moved the bin set appropriately, the offset parameters can be computed by solving

$$\begin{cases} s' = a + bs \\ e' = a + be \end{cases}$$

Hence $b = \dfrac{e' - s'}{e - s}$ and $a = s' - s\dfrac{e' - s'}{e - s}$.

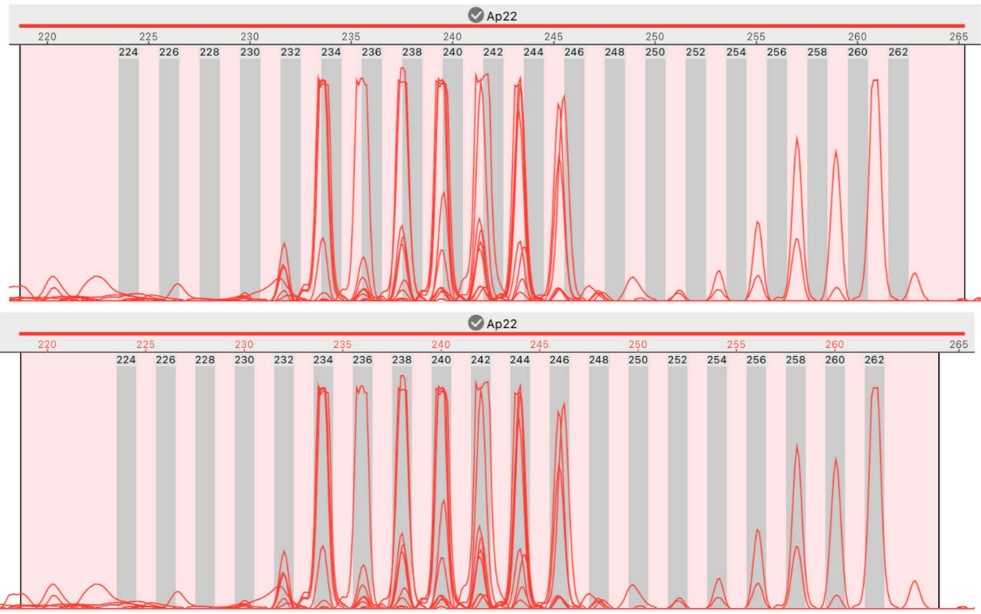

**Fig 3. A case of out-of-bin alleles that is solved.** Both images show the stacked traces from 8 samples of the same sequencer run. Peaks represent amplicons of a dinucleotide marker called "Ap22". Its range is represented by a horizontal red segment above the ruler showing graduations in base pairs (bp). Bins appear as grey rectangles. Top: peaks are shifted to the left with respect to bins, and more so for longer alleles, although bins are separated by exactly two base pairs. Bottom: the user has moved and narrowed the light-pink rectangle representing the range of the marker, such that bins coincide with peaks. This move translates into offset parameters $a = -6.40$ and $b = 1.029$ (see main text). As a result, the estimated size of peaks overlapping bin 258 (bottom image) has changed from ~257 bp to ~258.1 bp.

Since the user moves the bin set as a whole, the operation yields same offset parameters for all bins. These parameters are then associated to the chromatograms involved in the procedure (e.g., those displayed in Fig 3) and a given marker.

## Genotyping

In STRyper, a panel of microsatellite markers is associated to samples via dragging and dropping a panel icon onto the sample table (Fig 1) or via contextual menus, which initializes a genotype for each sample and each marker of the panel. Genotypes contain no allele information until alleles are called. Allele calling can be done manually by clicking peaks within a marker's range or automatically, via the implementation of a new algorithm.

This algorithm accounts for two main biochemical processes producing DNA fragments of different lengths for the same allele. One is the addition of a non-template nucleotide to the 3' end of the new DNA strand by the DNA polymerase during PCR [18]. Because the added nucleotide is generally an adenosine, this process is referred to as "adenylation". If adenylation affects only a portion of the amplicons, they may differ in length by one nucleotide, generating two peaks. The other process is "slippage" during replication, causing indels in the repeated region [19]. Slippage may result in a range of different amplicons that differ by the size of the repeat, a pattern known as "stuttering". These considerations served as a basis to develop a method for allele calling that first identifies peak clusters resulting from these processes (detailed in the S1 Text), and which accounts for the length of the repeat motive. In each delineated cluster, the most intense peak is considered as that representing the allele. Estimation of peak intensity accounts for clipping due to saturation of the fluorescence signal, in that the width of the saturated region is used when peak height/area may not reflect the quantity of

DNA material. Stuttering and adenylation are managed internally by the application, the user remains free to manually assign an allele to any peak.

Importantly, the method does not consider the absolute height or shape of a peak to call the first allele, beyond the fact that a minimal fluorescence level is required to delineate a peak (see S1 Text). If a peak is detected in the marker range (see below) and is not interpreted as crosstalk, at least one allele will be called. It was considered that the assessment of peak quality was better left to the user, who is expected to visually inspect every genotype.

For a diploid individual, the number of different alleles detected within a marker's range determines the individual's genotype: homozygous if one allele is detected, heterozygous otherwise. Because this inference is invalid for polyploid individuals, it was decided that only haploid and diploid markers could be defined in the application, constraining the maximum number or alleles per locus to 2. To cope with this constraint, the ability to annotate additional DNA fragments of interest, either automatically or manually, was implemented. Additional fragments may inform on the presence of paralogs, polyploidy, insufficient specificity of the PCR or contamination between samples. The application therefore distinguishes two types of peaks: those that are interpreted as alleles and whose number is limited to the ploidy of the marker, and others representing these additional DNA fragments. Because neither should comprise fragments produced by stuttering or adenylation, additional peaks are detected like alleles are (i.e., by identifying peak clusters). The relative height of peaks is used to categorize alleles (higher peaks) and additional peaks (smaller peaks).

All genotypes from samples of the current folder are listed in a table (Fig 4) that can be sorted and filtered according to various criteria (including allele names and sizes). This table lets users quickly scan genotypes, as corresponding peaks and allele labels of the selected genotype(s) appear on the right-pane. Correcting errors in allele call typically takes a single step: the user can simply drag the mouse from a peak to a bin, drag an allele label from one peak to another (Fig 5), or double-click a peak, which removes/attaches an allele from/to the peak. Double clicking allele labels lets users enter arbitrary allele names directly above peaks.

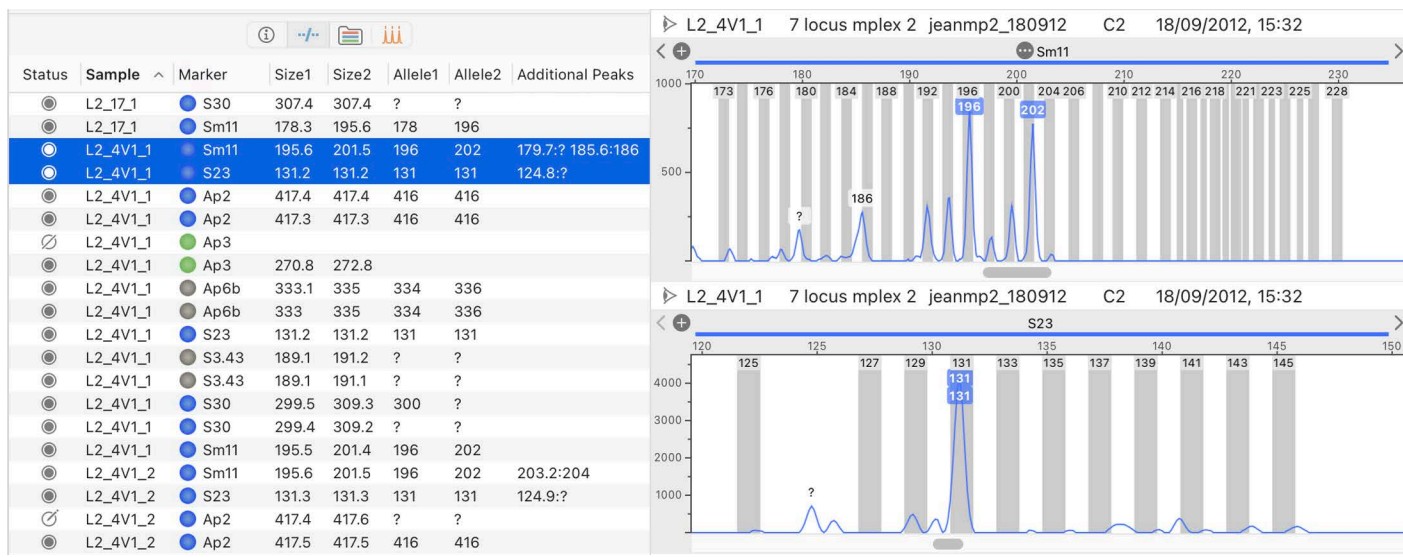

**Fig 4. The table listing genotypes in STRyper.** Each row represents a genotype at a unique sample and marker. Displayed traces in the right pane correspond to the selected genotypes. In this instance, two sections of the same trace are shown, corresponding to the range of each molecular marker. Peaks representing alleles are tagged with labels colored after the channel of the molecular marker. Labels with a white background represent additional peaks that were automatically flagged by the allele caller.

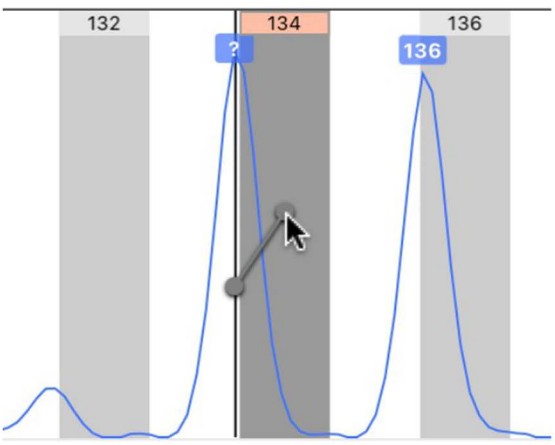 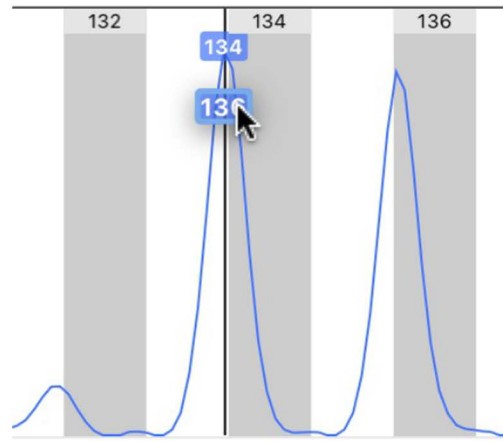

**Fig 5. Genotype editing by drag and drop in STRyper.** Vertical grey rectangles represent bins that define expected ranges of microsatellite alleles. Each bin has a name displayed on top. Allele names are represented by colored labels above peaks. Left-hand screen capture: the user is dragging the mouse from a peak to a bin. This will assign the peak an allele named after the bin, thereby replacing the question mark used for alleles that are out of bins. During the operation, a grey-colored handle connects the mouse location to another point horizontally located at the peak tip and vertically located at the clicked point. Right-hand screen capture: the user has decided that only the peak on the left should represent an allele and is dragging an allele label from the right-hand peak to the other. These actions are assisted by "magnetism" to lock the handle or allele label to the closest suitable destination, which triggers haptic feedback on the trackpad.

## Exporting results

STRyper allows exporting results in several ways. Genotypes and associated sample metadata can be exported as text files, or simply copied from selected table rows to a text editor or a spreadsheet application. In addition, a folder or a smart folder, with all its content – subfolders, samples, genotypes at microsatellite markers, associated marker panels (including bins) and custom size standards – can be archived and transferred between instances of the application. Upon importing an archived folder, any marker panel and size standard encoded in the archive is imported unless is it already in the database. The imported folder therefore shows the same content as the original.

## Evaluation of the software

### Usage and reliability

STRyper was developed to facilitate the genotyping of numerous individuals, from chromatogram import, management and viewing, to genotype editing and data export. How well it performs at these tasks cannot be evaluated without subjectivity.

The ability of an application to assign the right peaks to a DNA ladder fragments or alleles (i.e., allele calling) can be quantified more objectively by comparing these assignments to a reference, which is the assignments that an experienced user would have made by visually inspecting the chromatograms. Another reference, which could be used to evaluate the allele caller specifically, is the genotypes obtained at the same markers from an independent and more reliable method, typically amplicon sequencing. Such reference would allow detecting errors that even an experienced user would not detect. These errors may arise from variations in the motility of amplicons (leading to migration speed not being proportional to fragment length), due to the intrinsic properties of these fragments or variations in experimental conditions, including instruments and operators (reviewed in [20]). Mitigating errors that are

visually undetectable should not reasonably be expected from this application. Comparing genotypes called by STRyper to those obtained by sequencing would therefore not constituting a fair evaluation of the allele caller, even if sequence data were available for the same individuals and markers (I am not aware of the public availability of such dataset). The frequency of manual corrections that an experienced user must apply to automatic peak assignment was therefore used as a metric of the application performance, even though it is partly user dependent.

Given these limitations, it was considered more valuable to evaluate STRyper as part of an ongoing study (Vucić et al., in prep) instead of reanalyzing previously published data. Chromatograms were obtained from 314 individuals of the freshwater fish *Phoxinus lumaireul* (Teleostei, Cypriniformes), each amplified at two 6-plexes of microsatellite markers developed by Vucic *et al.* [21]. Amplicons were submitted to electrophoresis in an SeqStudio sequencer (Applied Biosystems) after addition of the GeneScan 500-LIZ size standard. After importing the 648 chromatograms (628 from amplified samples and 20 negative controls) into STRyper, the GeneScan 500 size standard was applied to each using the $3^{rd}$ degree polynomial as sizing method.

Ignoring electrophoresis failures that made 25 samples unusable, visual inspection of peak assignments to DNA ladder fragments revealed issues in eight chromatograms. In all cases, a size was not assigned to the appropriate peak or the DNA ladder because the peak was missing or abnormally short. As issues due to missing peaks cannot be fixed, manual corrections were applied to only four chromatograms. Overall, the verification of the DNA ladder for all chromatograms took less than five minutes.

For each marker, a set of bins was generated in one step by a specifying a bin width of 1 base pair and setting bin spacing according to the length of the microsatellite repeat motives [21]. For samples of a reference sequencing plate, the bin set was moved and resized as a whole, such that bins position matched peaks corresponding to alleles. For certain other sequencing plates and for five markers (PHOX4, PHOX11, PHOX29, PHOX33 and CtoA-247 [21]), peaks and bins appeared slightly misaligned (by less than 0.5 base pairs). Offset parameters for peak sizes were thus defined according to the procedure shown in Fig 3. This procedure made bins coincide neatly with peak locations for all regularly spaced alleles. I therefore saw no evidence that the use of linear relationship to estimate offset parameters was inappropriate. Individual bins were also added at locations indicating the presence of alleles that did not strictly follow the repeat pattern (probably due to mutations in microsatellite flanking regions).

Once these adjustments were done, genotypes were called and visually checked. Marker PHOX02 suffered from a combination of high stuttering, variable adenylation rates, the probable existence of mutations in flanking regions, which made peak assignment and binning very difficult, even visually. The marker was excluded, because if was considered too unreliable.

In several cases of PCR failures, the application assigned relatively faint peaks amounting to noise as alleles, which was expected. These cases were easily detected by visual inspection. The only common source of genotyping error resulted from varying degrees of adenylation at certain markers. The most intense peak or a cluster, which the application assigns to an allele, may sometime represent adenylated fragments and sometimes non-adenylated fragments. The estimated size of the same allele will therefore vary between individuals by approximately one base pair. This type of variation was much more rarely induced by stuttering, the degree of which is more constant.

In rare instances, peaks representing alleles had the same positions as taller peaks in other channels and were erroneously considered as resulting from crosstalk. These errors were

detected because neighboring peaks of similar shapes were present (indicating stuttering or heterozygosity) despite the absence of peaks in other channels at their position. More frequently, small artefactual peaks were not interpreted as crosstalk because their shape was irregular and/or their position was slightly shifted from that of the peaks that induced interference. This issue rarely affected genotyping as these peaks were generally too small to be considered as alleles. Rare errors occurred in very specific situations where the length of the alleles differed by only one base pair, such that the shorter peak was considered as the result of adenylation. Only visual comparison with other genotypes showed that adenylation was unlikely. The genotype caller does implement such check by comparing different genotypes called in the same batch (S1 Text), but this check may not always be effective. Finally, shorter allele dominance in heterozygotes [22], causing the peak representing the longer allele to be much smaller due to a very large difference in length between alleles ( > 60 bp), was not always properly managed. Admittedly, whether such peak should be considered as an allele is difficult to determine even for experienced users.

## Performance

During the evaluation, the performances of STRyper were monitored by debugging code and by the profiling tools of Xcode 15 on a MacBook Pro equipped with an M1 Pro chipset and a 120-Hz display comprising ~6M pixels. When it came to execution speed, importing the 648 chromatograms took 2.45 s, i.e., 264 chromatograms were imported per second on average. Application of the size standard (which involves peak assignment to DNA ladder fragment) took less than 0.12 s (~5400 chromatograms per second). Allele calling of the 3600 genotypes took 0.24 s (~15000 genotypes called per second). Since chromatograms/genotypes are processed successively in a single execution thread, the runtime of these tasks is proportional to the number of chromatograms or genotypes processed.

Memory usage was measured at 132 Megabytes (MB) after chromatogram import. It peaked at 250 MB after selecting the 3600 called genotypes and scrolling the 3600 traces from top to bottom and back. Memory usage peaked at 460 MB after selecting the 648 samples to display the stacked traces at the five channels (2000 traces displayed at once, as the application does not display more than 400 stacked traces per row).

All tasks other than those timed above were essentially instantaneous. Only the display of the 3600 genotypes and the 648 samples in the right pane induced a noticeable delay of about 1 second. Zooming and scrolling traces was generally achieved without noticeable frame drops, except when zooming in/out more than about 500 traces (stacked in several rows) near their full range (about 600 base pairs).

## Discussion

Based on its design, features and performance, STRyper should be a valuable tool for researchers who use traditional microsatellite markers. Genotyping hundreds of *Phoxinus* individuals at 12 markers with STRyper proved much faster than any of my previous genotyping jobs on similar data, keeping in mind that I cannot afford a comparison with recent versions of commercial competing applications. This test also showed that crosstalk detection and genotype calling was reasonably efficient, and could be improved upon. While the underlying methods can surely be refined, I believe that substantial improvements in these areas require comparisons between samples. Trained artificial intelligence has been proposed for the analysis of chromatograms [23], but this approach can only be used on limited set of microsatellite markers. As STRyper, nor any equivalent software, is not immune to genotyping errors (reviewed in [20]) one should always visually review genotypes and perform downstream corrections on exported results (e.g., [24–26]).

Independently of the performance of its allele caller, the main benefits of STRyper lie in its streamlined user interface that is optimized for the management and inspection of hundreds of chromatograms. This optimization is essential to population geneticists, who cannot spend as much time on individual genotypes as forensic researchers can. Since STRyper is not designed for diagnostics and must not be used for this task (it comes with no warranty), it does not assume that allele calls are reviewed by several users. Therefore, it does not record the history of manual corrections applied to genotypes (but still allows adding comments on genotypes). Such feature would have cluttered the user interface for very little benefits for most researchers.

Based on the reported metrics, users should not be concerned about the performance and responsiveness of STRyper. The size of the database and the number of samples contained in the selected folder should have little effect on the application performance and memory usage. The application essentially shows tables (including its right pane), for which only the visible rows, and a few others kept in cache for performance, are allocated in system memory (a feature provided by the NSTableView class of the AppKit framework). Rows that are not yet visible are not allocated, and those that move out view during scrolling eventually become deallocated. When chromatograms are fetched using textual metadata (sample name, plate well, plate name, run date, etc.), for example during a search though the whole database, only that piece of data is fetched from the store and allocated in memory (a feature of the Core Data framework). Fluorescence data is stored in separate objects (S1 Text) and is only fetched and allocated in memory when traces are displayed.

As the application only uses about 460 MB when displaying 2000 traces at once – the most that is allowed – memory usage should not be a concern either. The use of Apple-provided frameworks (mainly AppKit, Core Data and Core Animation) contributes to the low memory footprint and responsiveness of STRyper but would require a major rewrite of the GUI and database-management code if the application were to be ported to non-Apple platforms. However, methods related to chromatogram parsing, peak assignments (genotype calling and sizing) and drawing of fluorescence curves do not heavily depend on these frameworks (they mostly use functions written in plain C) and can be reused with only minor modifications.

From a GUI standpoint, several features of STRyper should be particularly useful to users. The first is the distinction between alleles and additional peaks. Since the number of peaks assigned to alleles never exceeds the marker ploidy, users should rarely need to remove peaks to correct a genotype that was called, a repetitive task that proved rather tedious in my previous genotyping jobs. The detection of additional peaks is optional, and these peaks can be reviewed, added manually, removed, or simply ignored as they are not part of an individual's genotype (they are listed and exported in a dedicated column). Theoretically, additional peaks should allow genotyping polyploid species, but I have not tested STRyper for this usage.

The second feature to underline is the implementation of fragment binning. The possibly to assign off-bin peaks to alleles (bins) via drag-and-drop (Fig 5, left) is certainly a time saver compared to typing allele names or selecting them among a long list. This task can even be avoided by minimizing the offset between peak and bin locations (Fig 3) prior to binning, in case variations in electrophoretic conditions have shifted the position of peaks relative to bins. This is currently done manually by the user, but a fully automatic, or user-assisted, procedure that minimizes the offset between peaks and bins (or theoretical fragment sizes) could be the goal of future developments. Granted, binning can be performed automatically by downstream programs like Tandem [17]. However, minimizing the offset between bins and peak representing "standard" alleles should help to distinguish alleles whose size do not follow the periodicity of the microsatellite repeat motive, and which may justify the creation of specific bins. Tandem alerts the user about problematic alleles but does not create new bins.

When it comes to database management, STRyper distinguishes itself by advanced search and filtering capabilities, which help reviewing problematic cases, among other benefits. For instance, all samples showing a particular allele at a marker can easily be retrieved across the whole database and displayed. To this end, samples can be gathered in a smart folder according to the name of the marker panel applied to them. Then, the list of their genotypes can be filtered based on the marker name, and the allele name or size. Any new genotyped sample presenting this allele would automatically appear in the smart folder.

Finally, the set of chromatograms contained in a folder (or a smart folder) with all its related data (marker panels and bin sets, custom size standard(s), genotypes…) is easy to share, as it can be transferred between instance of STRyper with a few mouse clicks and no option to set. Making folder archives available alongside any publication using STRyper should help to review results and to standardize the analysis of the same microsatellite markers by different researchers.

## Supporting information

**S1 Text. Details on peak detection and assignment, and on database management.** This file describes methods for peak delineation, baseline fluorescence level subtraction, determination of crosstalk, size assignment of molecular ladder fragments, detection of microsatellite alleles, and an overview of the database managed by STRyper.
(PDF)

**S1 File. Exported results of the analysis performed to evaluate the application.** The file contains a folder archive named "Phoxinus-2024.folderarchive". This archive contains all data related to the analysis of chromatograms from 34 *Phoxinus sp* individuals. The file can be imported in STRyper as a folder called "Phoxinus 2024". To do so, unzip the file if needed, and import "Phoxinus-2024.folderarchive" via the "File/Import Archived Folder…" menu of STRyper. See the STRyper help for more information.
(ZIP)

## Acknowledgements

I thank Dr. Douglass Hoffman from the United States National Institute of Health for his advice on decoding HID files, and numerous colleagues for testing STRyper. I also thank Drs. Matej Vucić and Frédéric Grandjean for giving access to *Phoxinus* chromatogram files, and Dr. Romain Pigeault for reading a draft of the manuscript.

## Author contributions

**Formal analysis:** Jean Peccoud.

**Funding acquisition:** Jean Peccoud.

**Methodology:** Jean Peccoud.

**Software:** Jean Peccoud.

**Writing – original draft:** Jean Peccoud.

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
