## [Decision Letter · Decision Letter 0]

27 Nov 2024

PONE-D-24-39975STRyper: a macOS application for microsatellite genotyping and chromatogram managementPLOS ONE

Dear Dr. Peccoud,

Thank you for submitting your manuscript to PLOS ONE. After careful consideration, we feel that it has merit but does not fully meet PLOS ONE’s publication criteria as it currently stands. Therefore, we invite you to submit a revised version of the manuscript that addresses the points raised during the review process.

We look forward to receiving your revised manuscript.

Kind regards,

Amod Kumar, Ph.D

Academic Editor

PLOS ONE

Journal Requirements: When submitting your revision, we need you to address these additional requirements. 1. Please ensure that your manuscript meets PLOS ONE's style requirements, including those for file naming. The PLOS ONE style templates can be found at https://journals.plos.org/plosone/s/file?id=wjVg/PLOSOne_formatting_sample_main_body.pdf and https://journals.plos.org/plosone/s/file?id=ba62/PLOSOne_formatting_sample_title_authors_affiliations.pdf 2. Thank you for stating the following financial disclosure: "Part of this work was funded by intramural funds from the CNRS and the University of Poitiers and by Agence Nationale de la Recherche grant ANR-20-CE02-0004 (SymChroSex)" Please state what role the funders took in the study.  If the funders had no role, please state: ""The funders had no role in study design, data collection and analysis, decision to publish, or preparation of the manuscript."" If this statement is not correct you must amend it as needed. Please include this amended Role of Funder statement in your cover letter; we will change the online submission form on your behalf. 3. Please review your reference list to ensure that it is complete and correct. If you have cited papers that have been retracted, please include the rationale for doing so in the manuscript text, or remove these references and replace them with relevant current references. Any changes to the reference list should be mentioned in the rebuttal letter that accompanies your revised manuscript. If you need to cite a retracted article, indicate the article’s retracted status in the References list and also include a citation and full reference for the retraction notice.

Reviewers' comments:

Reviewer's Responses to Questions

**Comments to the Author**

1. Is the manuscript technically sound, and do the data support the conclusions?

Reviewer #1: Yes

Reviewer #2: Yes

2. Has the statistical analysis been performed appropriately and rigorously? 

Reviewer #1: N/A

Reviewer #2: N/A

3. Have the authors made all data underlying the findings in their manuscript fully available?

Reviewer #1: Yes

Reviewer #2: Yes

4. Is the manuscript presented in an intelligible fashion and written in standard English?

Reviewer #1: Yes

Reviewer #2: Yes

5. Review Comments to the Author

Reviewer #1: Jean Peccoud's manuscript presents a software note on his application for microsatellite genotyping. For years, evolutionary biologists and ecologists working with microsatellite data have expressed frustration with available programs for analyzing capillary electrophoresis data—these programs are often prohibitively expensive, prone to bugs, slow, or have a steep learning curve. The most popular program, GeneMapper, embodies all these issues.

The release of STRyper is game-changing. It is free, fast, resource-efficient, and highly intuitive, enabling us to train students in its use far more quickly than with other programs. Its accuracy is very good, and its size-standard fitting, in my experience, outperforms that of GeneMapper.

While STRyper is currently exclusive to macOS, the author's rationale for this decision is clear. Given its lightweight nature and rapid performance, which does not necessitate a high-performance Mac workstation, I am confident that most labs would gain benefits from it.

This manuscript provides a comprehensive description of the software, leaving no room for further comments. I extend my sincere thanks to the authors for this invaluable contribution to our field.

H. Darras

Reviewer #2: This paper presents software programme for Apple Macintosh computers that can help researchers with genotype calling for microsatellites. The programme and its source code are freely available via GitHub. I have tried the programme and it is indeed easy to use with a very powerful GUI. It is great to see that the author created this software as a hobby, and despite the declining use of microsatellites, it will be very useful to evolutionary geneticist. The manuscript describing the software is generally well written and easy to read. However, I do have a number of criticisms, especially regarding the structuring of the text.

Main comments:

My main comment is that the manuscript is structured as a standard experimental paper with a methods section and a results section. This really does not fit the type of paper as it is simply a description of a software programme, so there are not really any methods and results. The paper therefore feels quite repetitive because some parts that logically should be combined are now split into two parts, one in the methods section and one in the results section. I would suggest to forego the methods/results structure in order to present the subjects in a more logical order. I suggest to start with description of the general characteristics of the programme, which is now in the beginning of the results section. The parts on evaluating the application can then be put in the end.

Small comments:

l. 37. It is not the motive itself that is mutating, but the number of times that the motive is repeated.

l. 39-45. I do not see why there are so many details about amplicon sequencing, as this is hardly relevant to the paper.

l. 54. hence -> therefore

l. 91. edition -> addition

l. 149. Can you give an explanation why the numbers 10 and 3 are used in this equation?

l. 162. whose -> that

l. 217-224. I found this description of the method difficult to follow, please revise.

l. 248-250. I did not understand this.

l. 268. Can you specify how the programme works with multiplexes?

l. 275. remove “when the project started”

l. 289. Amplicon sequencing can suffer from the same problems with null alleles as gel electrophoresis.

l. 304. Jelic -> et al.

l. 348. Rewrite “Most are at a couple of clicks away”

l. 391. what is meant with “original option”?

l. 400. remove “spontaneously”

l. 417-423. This part is especially repetitive as most of it is already written in the method section.

l. 447-448. What is meant with “issues in height chromatograms”?

l. 456. So what happened to this marker, was it excluded?

l. 481-482. This could be explained more clearly.

l. 484-485. I did not understand this.

l. 515. although improvable -> And could be improved upon.

l. 517-519. My guess is that this would be very difficult as you would need a very large set of high quality training data, where the genotypes have been scored correctly.

l. 519. Write “As STRyper, nor any other software, is not immune.

Some comments on the software itself (mere suggestions; these do not have to be implemented before the paper can be accepted for publication)

-I noticed that the export function in the file menu was always greyed out, also when the share icon at the bottom of the window was working.

-The binning menu was difficult to find, as it only appears when mousing over the name of the marker. Also, it was not self-evident that the little icon with the three dots was clickable.

-The left and right arrows in the chromatogram viewer (“move to next marker”) are not greyed out when there is no next marker to show. It therefore took me some time to discover what it does, as my first clicks did not do anything.

-The three icons (import, trash, filter) below the samples panel can be better placed in the window toolbar.

-The plus-icon at the top of the chromatogram viewer, was also quite confusing as it took me a while to figure out that you can only zoom in by dragging the mouse along the ruler on top of the viewer, which is easily missed as the target is very small. There also is no method for zooming out (I tried pressing the alt modifier key but that did not work).

-It would be nice if there could be an export function to export the called genotypes in a file format that is used by some popular software for population genetic analyses, such as GenAlex, Arlequin, Structure, or –because STRyper is a Mac only programme– GenoDive.

6. PLOS authors have the option to publish the peer review history of their article (what does this mean? ). If published, this will include your full peer review and any attached files.

**Do you want your identity to be public for this peer review?** For information about this choice, including consent withdrawal, please see our Privacy Policy .

Reviewer #1: No

Reviewer #2: No

---

## [Author Response · Author response to Decision Letter 1]

18 Dec 2024

Responses to reviewers appear in the file "response to reviewers"

---

## [Decision Letter · Decision Letter 1]

22 Jan 2025

STRyper: a macOS application for microsatellite genotyping and chromatogram management

PONE-D-24-39975R1

Dear Dr. Peccoud,

We’re pleased to inform you that your manuscript has been judged scientifically suitable for publication and will be formally accepted for publication once it meets all outstanding technical requirements.

Kind regards,

Amod Kumar, Ph.D

Academic Editor

PLOS ONE

Additional Editor Comments (optional):

I would like to congratulate for this wonderful work. Please do few minor changes as suggested by the reviewer.

Regards

Reviewers' comments:

Reviewer's Responses to Questions

**Comments to the Author**

1. If the authors have adequately addressed your comments raised in a previous round of review and you feel that this manuscript is now acceptable for publication, you may indicate that here to bypass the “Comments to the Author” section, enter your conflict of interest statement in the “Confidential to Editor” section, and submit your "Accept" recommendation.

Reviewer #2: All comments have been addressed

2. Is the manuscript technically sound, and do the data support the conclusions?

Reviewer #2: Yes

3. Has the statistical analysis been performed appropriately and rigorously? 

Reviewer #2: N/A

4. Have the authors made all data underlying the findings in their manuscript fully available?

Reviewer #2: Yes

5. Is the manuscript presented in an intelligible fashion and written in standard English?

Reviewer #2: Yes

6. Review Comments to the Author

Reviewer #2: The author did a great job in revising the manuscript. Freeing the manuscript from the standard “Methods - Results” structure has really resulted in a text that is much easier to follow and much clearer. This makes it stand out better what a useful tool he has created with STRyper. I have no further large comments, but there is a small number of writing issues, that should only take about tens minutes to correct.

l. 34 for -> from

l. 77. in -> for

l.113. expect -> except

l. 186. statured -> saturated

l. 230. The poor score -> A poor score

l. 244. Fix the “Fig2Error!”

l. 385. Exportation of -> Exporting

l. 386. exportation -> exporting

l. 397. genotyping -> genotype

l. 398. remove “a part of”

l. 437. described -> shown

l. 509. guaranty -> warranty

7. PLOS authors have the option to publish the peer review history of their article (what does this mean? ). If published, this will include your full peer review and any attached files.

**Do you want your identity to be public for this peer review?** For information about this choice, including consent withdrawal, please see our Privacy Policy .

Reviewer #2: No

---

## [Editor Report · Acceptance letter]

PONE-D-24-39975R1

PLOS ONE

Dear Dr. Peccoud,

I'm pleased to inform you that your manuscript has been deemed suitable for publication in PLOS ONE. Congratulations! Your manuscript is now being handed over to our production team.

Kind regards,

on behalf of

Dr. Amod Kumar

Academic Editor

PLOS ONE